# LoFT: Finding Lottery Tickets through Filter-wise Training

**Qihan Wang⋆, Chen Dun⋆, Fangshuo Liao⋆, Chris Jermaine, Anastasios Kyrllidis**
Department of Computer Science
Rice University

## Abstract

In this paper, we explore how one can efficiently identify the emergence of "winning tickets" using distributed training techniques, and use this observation to design efficient pretraining algorithms. Our focus in this work is on convolutional neural networks (CNNs), which are more complex than simple multi-layer perceptrons, but simple enough to exposure our ideas. To identify good filters within winning tickets, we propose a novel filter distance metric that well-represents the model convergence, without the need to know the true winning ticket or fully training the model. Our filter analysis behaves consistently with recent findings of neural network learning dynamics. Motivated by such analysis, we present the *LOttery ticket through Filter-wise Training* algorithm, dubbed as LoFT. LoFT is a model-parallel pretraining algorithm that partitions convolutional layers in CNNs by filters to train them independently on different distributed workers, leading to reduced memory and communication costs during pretraining. Experiments show that LoFT $i$) preserves and finds good lottery tickets, while $ii$) it achieves non-trivial savings in computation and communication, and maintains comparable or even better accuracy than other pretraining methods.

## 1 Introduction

The Lottery Ticket Hypothesis (LTH) [8] claims that neural networks (NNs) contain subnetworks ("*winning tickets*") that can match the dense network's performance when fine-tuned in isolation. Yet, identifying such subnetworks often requires proper pretraining of the dense network. How to efficiently find such subnetworks remains a widely open question: since LTH relies on a *pretraining* phase, it is a *de facto* criticism that finding such pretrained models could be a burdensome task, especially when one focuses on large NNs.

This burden has been eased with efficient training methodologies, which are often intertwined with pruning steps. Simply put, one has to answer two fundamental questions: *"When to prune?"* and *"How to pretrain such large models?"*. Focusing on *"When to prune?"*, one can prune before [23, 22, 35], after [21, 13, 7, 12, 24, 29, 11, 36, 41], and/or during pretraining [8, 33, 27, 2, 6, 30, 28].[1] Works like SNIP [23, 22] and GraSP [35] aim to prune without pretraining, while suffering some accuracy loss. Pruning after training often leads to favorable accuracy, with the expense of fully training a large model. A compromise between the two approaches exist in *early bird tickets* [39], where one could potentially avoid the full pretraining cost, but still identify "winning tickets", by performing a smaller number of training epochs and lowering the precision of computations. *This finding reveals the opportunity to design more efficient pretraining algorithms that target specifically at identifying the winning tickets for larget models.*

---

[1]LTH approaches, while they originally imply pruning after training, include pruning at various stages during pretraining to find the sparse subnetworks.

Has it Trained Yet? Workshop at the Conference on Neural Information Processing Systems (NeurIPS 2022).

Focusing on *"How to pretrain large models?"*, modern large-scale neural networks come with significant computational and memory costs. Researchers often turn to distributed training methods, such as data parallel and model parallel [43, 1, 34, 3, 42, 9, 10, 4], to enable heavy pretraining towards finding winning tickets, by using clusters of compute nodes. Yet, data parallelism needs to update the whole model on each worker—which still results in a large memory and computational cost. To handle such cases, researchers utilize model parallelism, such as Gpipe [17], to reduce the per node computational burden. Traditional model parallelism enjoys similar convergence behaviour as centralized training, but needs to synchronize at every training iteration to exchange intermediate activations and gradient information between workers, thus often incurring high communication cost.

**Our approach and contributions.** In this work, we propose a new model-/data-parallel pretraining method on the one-shot pruning setting that can efficiently reveal winning tickets for convolutional neural networks (CNNs). In particular, we center on the following questions:

*"What is a characteristic of a good pretrained CNN that contains the winning ticket? How will such a criterion inform our design towards efficient pretraining?"*

Prior works show that filter-wise pruning is more preferable compared with weight pruning for CNNs [17, 16, 5, 37, 25]. Our approach operates by decomposing the full network into narrow subnetworks via filter-wise partition during pretraining. These subnetworks, which are randomly recreated intermittently during the pretraining process, are trained independently and their updates are periodically aggregated into the global model. Because each subnetwork is much smaller than the full model, our approach also enables scaling beyond the memory limit of a single GPU. Our methodology allows the discovery of winning tickets with less memory and a lower communication budget. The contributions of our work are summarized as follows:

- We propose a metric to quantify the distance between tickets in different stages of pretraining, allowing us to characterize the convergence to winning tickets throughout the pretraining process.

- We identify that such convergence behavior suggests an alternative way of pretraining: we propose a novel data-/model-parallel pretraining method through filter-wise partition of CNNs and iterative training of such subnetworks.

- We theoretically show that our proposed method achieves CNN weight that is close to the weight found by gradient descent in a simplified scenario.

- We empirically show that our method provides a better or comparable winning ticket, while being memory and communication efficient.

## 2 Identifying Winning Tickets Early in the Training Process

The CNN model [14, 19, 31] is composed of convolutional layers, batch norm layers [18], pooling layers, and a final linear classifier layer. Our goal is to retrieve a *structured* winning ticket, through partitioning and pruning the filters in the convolutional layers. Let $p_i$ denote the number of input channels for the $i$-th convolutional layer. Correspondingly, the output channel of the $i$-th layer is the same as the input channel of the $(i+1)$-th layer, which is $p_{i+1}$. Let $h_i, w_i$ be the height and width of the input feature maps, respectively. Then, the $i$-th convolutional layer transforms the input feature map $x_i \in \mathbb{R}^{p_i \times h_i \times w_i}$ into the output feature map $x_{i+1} \in \mathbb{R}^{p_{i+1} \times h_{i+1} \times w_{i+1}}$ by performing 2D convolutions on the input feature map with $p_{i+1}$ filters of size $3 \times 3$, where the $j$-th filter is denoted as $\mathcal{F}_{i,j} \in \mathbb{R}^{p_i \times 3 \times 3}$. Thus the total filter weight for the $i$-th layer is $\mathcal{F}_i \in \mathbb{R}^{p_{i+1} \times p_i \times 3 \times 3}$. Formally, pruning $1/k$ of the filters in the $i$-th layer is equivalent to discarding $p_{i+1}/k$ filters. Thus the resulted total pruned filter weight is in $\mathbb{R}^{p_{i+1} \cdot (k-1)/k \times p_i \times 3 \times 3}$ and the output feature map $x_{i+1}$ is in $\mathbb{R}^{p_{i+1} \cdot (k-1)/k \times h_{i+1} \times w_{i+1}}$.

### 2.1 Evaluate the distance of two pretrained models

Borrowing techniques from search system rankings [20], we propose a *filter distance* metric, based on a position-weighted version of Spearman's footrule [32]. In particular, consider the filters at epochs $X$ and $Y$ on the $i$-th layer, namely $\mathcal{F}_i^{(X)}, \mathcal{F}_i^{(Y)}$. We calculate the $\ell_2$-norm of $\mathcal{F}_{i,j}^{(X)}, \mathcal{F}_{i,j}^{(Y)}$ for each filter index $j \in [p_{i+1}]$ and sort them by magnitude. We denote the two sorted list with length $p_{i+1}$ as $R^{(X)}$ and $R^{(Y)}$, with each containing the $\ell_2$-norm of the filters, $\left\| \mathcal{F}_{i,j}^{(X)} \right\|_2$ and $\left\| \mathcal{F}_{i,j}^{(Y)} \right\|_2$.

We represent the change in ranking from $R^{(X)}$ to $R^{(Y)}$ as $\sigma$. Namely, if $x \in R^{(X)}$ is the $i$-th element in $R^{(X)}$, then, the ranking of $x$ in $R^{(Y)}$ is denoted as $\sigma(i)$. The original Spearman's footrule defines the displacement of element $i$ as $|i - \sigma(i)|$, leading to the total displacement of all elements as:

$$F(\sigma) = \sum_i |i - \sigma(i)|.$$

Given weights $w_i$'s for the elements, the weighted displacement for element $i$ becomes $w_i \cdot \left| \sum_{j<i} w_j - \sum_{\sigma(j)<\sigma(i)} w_j \right|$, leading to the total weighted displacement as follows: [2]

$$F_w(\sigma) = \sum_i w_i \cdot (| \sum_{j<i} w_j - \sum_{\sigma(j)<\sigma(i)} w_j |).$$

To put emphasis on the correct ranking of the top elements, we set the position weight for the $i$-th ranking element as $1/i$. To further simplify calculations, we approximate $\sum_{i=1}^n \frac{1}{i} \approx \ln(n) - \ln(1)$ where $\ln(\cdot)$ is the natural logarithm. The above lead to the following definition for our *filter distance*:

$$F_{\text{filter}}(\sigma) = \sum_i \tfrac{1}{i} \cdot |\ln(i) - \ln(\sigma(i))|.$$

For the case where the two lists of pruned filters do not contain the same elements, we can naturally define the distance when the $i$-th element is not in the other list to be $|\ln(l+1) - \ln(i)|$; $l$ is the length of the pruned filter list. This filter distance metric is fundamentally different from the mask distance proposed in [38]. We compare against these early-pruning methods in the experiments.

## 2.2 Rethinking the Property of Winning Tickets

Through empirical analysis, we observe that *training the CNN weights until loss converges is not necessary for the discovery of winning tickets.* However, many existing pretraining algorithms do not exclude heavy training over the whole CNN model. *These facts demand a new pretraining algorithm, targeting specifically at efficiently finding winning tickets.*

We propose sampling and training different sets of tickets during different stages of the pretraining. In this way, the algorithm is expected to "touch" upon the potential winning tickets at certain iterations. We conjecture (this is empirically shown in our experiments) that important filters in such winning tickets can be preserved and further recovered at the end of pretraining using our approach. These observations led us to the definition of the LoFT algorithm.

## 3 LoFT: Finding Lottery Tickets Through Filter-wise Training

**Algorithm 1** LoFT Algorithm

---
1: **Parameter**: $T$ synchronization iterations in pretraining, $S$ workers, $\ell$ local iterations, $W$ CNN weights,

---
2: $h(W) \leftarrow$ randomly initialized CNN.
3: **for** $t = 0, \ldots, T-1$ **do**
4:    $\{h_s(W_s)\}_{s=1}^S = \texttt{filterPartition}(h(W), S)$
5:    Distribute each $h_s(W_s)$ to a different worker.
6:    **for** $s = 1, \ldots, S$ **do**
7:       Train $h_s(W_s)$ for $\ell$ iterations using local SGD.
8:    **end for**
9:    $h(W) = \texttt{aggregate}\left(\{h_s(W_s)\}_{s=1}^S\right).$
10: **end for**

---

We treat "*sampling and training sets of tickets*" as a filter-wise decomposition of a given CNN, where each ticket is a subnetwork with a subset of filters. The LoFT algorithm that implements our ideas is shown in Algorithm 1. Each block within a CNN typically consists of two identical convolutional layers, $\text{conv}_i$ and $\text{conv}_{i+1}$. Our methodology operates by partitioning the filters of these layers, $\mathcal{F}_i$ and $\mathcal{F}_{i+1}$, to different subnetworks –see $\texttt{filterPartition()}$ step in Algorithm 1– in a structured, disjoint manner. These subnetworks are trained independently –see local SGD steps in Algorithm 1– before aggregating their updates into the global model by directly placing the filters back to their original place—see $\texttt{aggreegate()}$ step in Algorithm 1. *The full CNN is never trained directly.*

Our methodology of choosing tickets/subnetworks avoids partitioning layers that are known to be most sensitive to pruning, such as strided convolutional blocks [26]. Parameters that are not

---

[2]Using other norm calculation like $\ell_2$-norm will not affect the overall characteristic of filter distance in analysis.

partitioned are shared among subnetworks, so their values must be averaged when the updates of tickets/subnetworks are aggregated into the global model.

Compared with common distributed protocols, our pretraining methodology $i$) reduces the communication costs, since we only communicate the tickets/subnetworks; and $ii$) reduces the computational and memory costs on each worker, since we only locally train the sampled tickets/subnetworks that are smaller than the global model. From a different pespective, our approach allows pretraining networks beyond the capacity of a single-GPU: The global model could be a factor of $O(S)$ wider than each subnetwork, allowing the global model size to be extended far beyond the capacity of single GPU. *The ability to train such "ultra-wide" models is quite promising for pruning purposes.*

After pretraining with LoFT, we perform standard pruning on the whole network to recover the winning ticket, and use standard training techniques over this winning ticket until the end of training.

**Theoretical result.** We perform theoretical analysis on a one-hidden-layer CNN, and show that *the trajectory of the neural network weight in* LoFT *stays near to the trajectory of gradient descent.*

**Theorem 1.** *Let $f(\cdot, \cdot)$ be a one-hidden-layer CNN with the second layer weight fixed. Assume the number of hidden neurons satisfies $m = \Omega\left(\left(n^4 K^2 / \lambda_0^4 \delta^2\right) \cdot \max\{n, d\}\right)$ and the step size satisfies $\eta = O\left(\frac{\lambda_0}{n^2}\right)$: Then, under mild assumptions, with probability at least $1 - O(\delta)$ we have:*

$$\mathbb{E}_{[\mathbf{M}_T]}\left[\left\|\mathbf{W}_T - \hat{\mathbf{W}}_T\right\|_F^2\right] + \eta \sum_{t=0}^{T-1} \mathbb{E}_{[\mathbf{M}_T]}\left[\left\|f(\mathbf{X}, \mathbf{W}_t) - f\left(\mathbf{X}, \hat{\mathbf{W}}_t\right)\right\|_2^2\right] \leq O\left(\frac{n^2 \sqrt{d}}{\lambda_0^2 \kappa m^{\frac{1}{4}} \sqrt{\delta}} + \frac{2\eta^2 T \theta^2 (1-\xi)\lambda_0}{S}\right).$$

**Remarks.** Intuitively, this theorem states that the sum of the expected weight difference in the $T$-th iteration (i.e., $\mathbb{E}_{[\mathbf{M}_T]}[\|\mathbf{W}_T - \hat{\mathbf{W}}_T\|_F^2]$) and the aggregation of the step-wise difference of the neural network output between LoFT and gradient descent (i.e., $\sum_{t=0}^{T-1} \mathbb{E}_{[\mathbf{M}_T]}\|f(\mathbf{X}, \mathbf{W}_t) - f\left(\mathbf{X}, \hat{\mathbf{W}}_t\right)\|_2^2$) is bounded and controlled by the quantity on the right-hand side. I.e., both the weights found by LoFT as well as the output of LoFT are close to the ones found by regular training. Notice that increasing the number of filters $m$ and the number of subnetworks $S$ will drive the bound of the summation to zero.

| SETTING | NO-PRUNE | METHODS | PRUNING RATIO | | | COMM COST | IMPROV. |
|---|---|---|---|---|---|---|---|
| | | | 80% | 50% | 30% | | |
| RESNET34 CIFAR-100 | 75.93 | GPIPE-2 | 75.51 | 76.00 | | 131.88G | |
| | | LoFT-2 | 76.11 | 77.07 | | 104.78G | 1.26× |
| | | GPIPE-4 | 75.51 | 76.00 | | 461.60G | |
| | | LoFT-4 | 75.05 | 76.51 | | 144.66G | 3.19× |
| PRERESNET-18 IMAGENET | 70.71 | GPIPE-2 | 66.71 | 69.14 | 70.29 | 20954.24G | |
| | | LoFT-2 | 65.41 | 69.12 | 69.64 | 791.09G | 21.60× |
| | | GPIPE-4 | 66.71 | 69.14 | 70.29 | 52385.59G | |
| | | LoFT-4 | 65.60 | 68.93 | 69.77 | 1284.84G | 40.77× |

Table 1: Left: Fine-tuned accuracy for different pretraining methods at different pruning ratios. NO-PRUNE corresponds to full CNN training without pruning. Right: Total communication costs (COMM) of model parallel baseline (GPipe) [17] and LoFT during pretraining. Number after method name represents the number of parallel worker used.

## 4 Experiments

We show that LoFT can preserve the winning tickets and non-trivially reduce costs during pretraining. We illustrate that LoFT does not recover the winning tickets by chance: LoFT converges to winning tickets faster and provide better tickets for all pretraining length.

**Experimental Setup.** For each setting, we consider a workflow where we pre-train for 20 epochs and then fine-tune for 90 epochs. We consider 3 networks: PreActResNet-18, PreActResNet-34 [15], and WideResNet-34 [40] to characterize our performance on models of different sizes and structures. We test these settings on the CIFAR-100, and ImageNet.

Table 1 shows the performance comparison for LoFT and Gpipe under various settings. While LoFT inherits the memory efficiency from model-parallel training methods, it further reduces the communication cost from $1.26\times$ up to $40.77\times$, as shown in Table 1. This is achieved by $i$) changing

the way of decomposing the network such that each worker can host an independent subnetwork and train locally without communication, which greatly reduces the communication frequency; and $ii)$ each worker only exchange the weight of the subnetwork after each round of local training instead of transmitting activation maps and gradients.

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
