# OpenReview forum: "LOFT: Finding Lottery Tickets through Filter-wise Training"
_NeurIPS.cc/2022/Workshop/HITY — HITY Workshop NeurIPS 2022_

### Official Review · Reviewer_5nvL · 2022-10-10
**Filter-wise model-parallel training**

**Rating:** 0
**Confidence:** 2

**Review:**

The paper introduces a different way to perform model-parallel training by splitting the model filter-wise. Effectively, each worker is training a subnetwork only, which leads to reduced communication costs.

My main critique is that the algorithmic details and the experimental results are unclear for the current description (see below). Furthermore, I am not sure I see the connection between the proposed method and the Lottery Ticket Hypothesis. LoFT doesn't "find lottery tickets" but is an alternative model-parallel training strategy.
Due to these reasons, I vote to reject the paper. However, I have to admit that I am not too familiar with the literature in this area. If other reviewers accept the paper, I don't have an issue with this decision.

Feedback:
- The intro part of Section 2 (Lines 67-78) introduces a lot of notation that isn't ultimately necessary for the rest of the paper. Perhaps this section could be significantly compressed.
- I did not understand how/why the weighted displacement was introduced (Lines 88-89). However, I am also not familiar with the surrounding literature.
- Algorithm 1 is hard to understand. Line 2 introduces h(W), which is probably the model's forward function? The function `filterPartition` seems to be really crucial for this algorithm but is never explained in any detail.
- The paragraph starting at Line 126 mentions that some parameters/layers are not partitioned. Could you provide some more information about this, e.g., what is the ratio of these parameters compared to the entire network? How much manual work is required here? Is this algorithm then applicable to other problems, e.g. NLP tasks (or other tasks where there is no prior work on layers "that are known to be most sensitive to pruning")?
- Theorem 1 uses notation that was never really introduced (or I missed it). E.g., $M_T$, $K$, etc.
- There is no proof provided for Theorem 1.
- The paper is missing some crucial experimental details. Describing them would help the readability of the paper. Some examples that were unclear to me:
  - For the experiments, you train for 20 epochs, then prune, and then train for another 90 epochs, right? Only the last 90 epochs are then trained with LOFT/GPipe?
  - The text mentions that 3 networks were considered, PreActResNet-18, PreActResNet-34, and WideResNet-34. Table 1 mentions ResNet34 and PreActResNet-18. Where are the results from the WideResNet-34?
  - Line 154 mentions LoFT provides "better tickets for all pretraining lenght[s]". But for CIFAR-100, LoFT beats GPipe only in 3/4 cases (would be totally fine) and for ImageNet in 0/6 cases. Here, I am comparing the numbers from GPipe and LoFT in Table 1. It seems that LoFT has a significantly reduced communication need, but then the claim shouldn't be that it provides "better tickets".

Nits:
- You could sort and compress the citations, e.g. Line 27 could be "[22,23,25], after [7, 11-13, 21, 24, 29, 36, 41]..."
- Line 5: "enough to [expose] our ideas".
- Line 33: "pretraining algorithms that [are] specifically [targeted] to identify the winning tickets for large[t] models."
- Line 34: I don't understand what is meant by "heavy" pretraining.
- Line 62: "CNN weight" -> "CNN parameters"
- Line 67: "The CNN model" -> "A typical CNN model"
- Inconsistent use of hyphens, e.g. Line 123 vs. 124.
- Line 154: "and provide[s] better winning tickets for all pretraining lenght[s]".

---

### Official Review · Reviewer_d3CS · 2022-10-18
**An efficient and well-motivated approach to find early LTs leveraging model and data parallelism**

**Rating:** 1
**Confidence:** 4

**Review:**

This paper proposes a novel method to early identify lottery tickets, leveraging distributed computation. The background review is exhaustive and the motivation is compelling. The proposed method includes a novel and efficient metric for convolutional filter distances that empirically seems to expose lottery tickets long before network convergence. The method leverages distributed computation achieving substantial speedup over other relevant parallelization methods, and the full network is never trained completely, providing scalability also in terms of memory. Theoretically, a result is presented (with no proof) by which the method is stable and more performative compared to full SGD. Despite being conceptually limited to CNNs (and experimentally to vision discriminative tasks), this is a valuable and promising contribution that I propose to accept.

Questions/Comments:
* As a comment for interesting future work, the following property of the algorithm deserves attention: partitions contain disjoint filters (line 122), and parameters that are not partitioned are shared and their values "must be averaged" (line 128). I think this is a non-trivial step and some interesting implicit biases are at work. What is the difference in behaviour between disjoint and averaged partitions? When is it generally best to use one or the another?
* As currently presented, Theorem 1 is hard to grasp, despite the intuitions provided. I recommend to add an appendix with proof derivation and rationale. Also, it is not 100% clear to me whether te provided bound determines that LoFT training is "close" to full SGD in any meaningful way, given that the Frobenius/L2 norm may not be the most accurate or relevant one in terms of information and/or performance.

---

### Decision · Program_Chairs · 2022-10-20

Accept